# Blended Natural Support Materials—Collagen Based Hydrogels Used in Biomedicine

**DOI:** 10.3390/ma13245641

**Published:** 2020-12-10

**Authors:** Ruxandra-Elena Geanaliu-Nicolae, Ecaterina Andronescu

**Affiliations:** Department of Science and Engineering of Oxide Materials and Nanomaterials, Faculty of Applied Chemistry and Materials Science, University Politehnica of Bucharest, 060042 Bucharest, Romania; ecaterina.andronescu@upb.ro

**Keywords:** collagen, hydrogels, biomedicine, drug delivery systems, wound healing, tissue engineering

## Abstract

Due to their unique properties—the are biocompatible, easily accessible, and inexpensive with programmable properties—biopolymers are used in pharmaceutical and biomedical research, as well as in cosmetics and food. Collagen is one of the most-used biomaterials in biomedicine, being the most abundant protein in animals with a triple helices structure, biocompatible, biomimetic, biodegradable, and hemostatic. Its disadvantages are its poor mechanical and thermal properties and enzymatic degradation. In order to solve this problem and to use its benefits, collagen can be used blended with other biomaterials such as alginate, chitosan, and cellulose. The purpose of this review article is to offer a brief paper with updated information on blended collagen-based formulations and their potential application in biomedicine.

## 1. Introduction

In the 1960s and 1970s laboratories started to focus on collagen for medical application of biomaterials and connective tissue. At the same time, the technology helped researchers to more easily obtain the materials and to start controlling their properties according to the desired potential application. Biopolymers offer expanded opportunities in biomedical applications by enabling modulation of their properties such as cell adhesion, migration, proliferation, and incorporation of different types of drugs in correlation with the chemical structure compatibility [1,2].

Collagen is one of the most studied biopolymers in medicine: Over 260,000 literature articles mention it as key element in tissue regeneration, being surnamed “the steel of the biological material” [3]. The effort of scientists to know its implications in various normal or pathological processes in the body is explained by the ability to interact with a large number of molecules that can trigger biological changes, being the main biopolymer in the extracellular matrix of vertebrates and invertebrates, the substance that keeps the body together [4,5,6]. One of the factors that recommends it for a wide range of uses including medicine is its versatility, the ability to be used in various formulations and to blend with other compatible biomaterials such as alginate, chitosan, hyaluronic acid, cellulose. Having the classic properties of biomaterials (biocompatibility, biodegradability, and biomimetic) blended materials benefit from the properties of each of the compounds, but also from the interaction between them that leads to the improvement of each properties. It is well known that inside the body collagen has good mechanical stability, elasticity, and thermal and enzymatic stability, but when it is extracted and used, these properties are much reduced, needing to be improved for possible applications in biomedicine. This is explained by the fact that the most-used type of collagen is type I collagen, although the good qualities mentioned previously are present in all types of collagen [7,8,9].

Its disadvantages (mechanical strength, thermal stability, and enzymatic degradation) can be improved using different methods: Crosslinking (chemical, physic, and enzymatic), covalent conjugating, grafting polymerization, or blending. In addition, collagen has low immunogenicity, good in vivo absorption, and synergism with other bioactive compounds and haemostatic properties. However, blended collagen-based biomaterials are a method of improving collagen disadvantages and using its benefits in different physical forms such as films, microspheres, micro and nanoparticles, and coatings [10,11,12]. Hydrogels are among the most important structures used in drug delivery systems, having the ability to swell and absorb biological fluids, with a stable structure that remains unchanged after many processes. Controlled release systems require specific properties of biomaterials such as mechanical, pH, enzymatic, and thermal stability, properties not held by collagen but which are obtained in combination with chitosan or alginate for the release of analgesics, chemotherapeutics, and even natural active principles such as aloe vera or curcumin. In the optical biomedical field, blended collagen hydrogels with chitosan or alginate are used for corneal disease with good mechanical and optical properties, and transparency. For dressings it is necessary to keep the wound moist, and for low adhesion, adsorbed blood and excess exudate, protection against infections, and permeability. These properties have been obtained for several forms of blended collagen-based materials: Films, microspheres, membranes, and scaffolds. These all formulations can be used for other parts of the body to benefit from treatment based on blended collagen-based materials when damaged [7,8,9,10,11,12,13,14,15].

Since 1997, the researchers’ interest in collagen and blended collagen-based biomaterials has been increasing according to a statistical search from Science direct database, using the keywords blended collagen and collagen. It was found that the annual number of articles increased continually from 141 in 1997 to 1763 in 2020 for blended collagen biomaterials (Figure 1). On the other hand, different commercial blended collagen-based products are on the market and used for tissue repair. This indicates the importance and potential use in biomedicine for biomaterials.

Until now, existing reviews focused on other methods of improving collagen properties, mostly chemical and physical crosslinking of collagen-based materials and their potential applications. Even if there is a multitude of formulations and potential applications for blended collagen-based biomaterials, it is found that a quantitative comparison of the obtaining parameters cannot be made, with each research group choosing its own synthesis protocol. Thus, the comparison that can be made is regarding the synthesis route as a whole, qualitative characteristics, and the potential application. This review aims to provide a guide of the most relevant studies for design of future experiments on collagen-based blended biomaterials. This paper reviews the latest collagen-based formulations in order to evaluate the state of recent research for potential biomedical application. The paper also includes discussions of structure, synthesis, characterization, and properties of the presented formulation based on collagen–chitosan, collagen–alginate, collagen–cellulose, or other blended collagen-based biomaterials.

## 2. Biopolymers Structure and Properties

Biopolymers are one of the most iconic classes of soft materials, smart and suitable materials for various applications. The most-used formulations of blended biopolymers are hydrogels, films, microspheres, nanofibers, and sponge. Hydrogels are three-dimensional crosslinked networks of polymers which can absorb and retain significant amount of water, without dissolving or losing their three-dimensional structure. Hydrogels can be synthetic or natural materials, with different chemical composition, mechanical, physical and chemical properties in correlation with the application. There are two types of polymers that form hydrogels: Hydrophilic and hydrophobic. Hydrophilic polymers with functional groups hydroxyl (-OH), amine (NH_2_), or amide (-CONH-CONH_2_), have the ability to absorb water and expand the hydrogel structure (swelling process). The hydrophobic polymers have lower swelling capacity, but present significantly enhanced strength, surface hydrophobicity and antidrying, despite their extremely high water content [13,14,15]. 

A frequent classification for hydrogels is according to the type of crosslinking strategy governing the polymer chain interactions because the interactions inside the hydrogel network define the network structure and hydrogel properties, including mechanical strength, a very important property in biomedical applications field [16]. 

Microspheres are spherical particles with a large cell attachment surface mostly used as drug delivery systems for drugs, cells, and biological tissue regenerations factors. Generally, the most-used methods of synthesis are direct aliquoting and emulsification, which have the disadvantages of non-uniform dimensions and shape for obtained microspheres. Recently, these methods were improved through the application of microfluid technology and using chip strategy, step by step [17,18,19].

Scaffolds are collagen sponges with 3D structure. In literature are used different synthesis methods: Freeze drying, electrospinning, and 3D printing. Due to its benefits of maintaining the initial structure and biological properties of collagen and active principle loaded, freeze drying is the most-used method. However, it is not resource, cost, and time efficient. Although wet electrospinning is environmentally friendly, both electrospinning methods affect the native structure of collagen and active substances loaded [20,21]. When using 3D printing, collagen does not have the necessary accuracy but this method can be combined with crosslinking and freeze drying when improved structure and mechanical properties are obtained. Even if microphysiological devices, patterned tissues, perfusable vascular-like networks and implantable scaffolds were developed, printing dynamic biological materials to reproduce patient-specific anatomical structure is difficult. Collagen 3D printing in its native unmodified form is a real challenge for researchers because gelation is typically achieved using thermally driven self-assembly [22,23].

Films are thin and flexible layers of polymer, interesting for researchers due to their unique nanoscale characteristics which increase patient compliance. Films can be used as drug release systems; however, their limits are limited capacity to load high dosage or many drugs and quick drying time (1 h at room temperature). Still, films can target sensitive places which are not feasible for other formulations [24].

Comparing all methods that improve biomaterials properties, the most-used is chemical crosslinking with synthetic agents. Even if this method is quick and easy to use, and also not expensive, it presents a major disadvantage: Side effects given by the presence of crosslinking agent (such as glutaraldehyde-GA, formaldehyde-FA, 1-ethyl-3-(3-dimethyl aminopropyl) carbodiimide- EDC, N-hydroxysuccinimide-NHS, genipin, ans proanthocyanidin) after washing the materials. Crosslinking being a post-synthesis applied method onto the material surface, produces non-uniform structure. First used crosslinking agent was GA due to its high reactivity availability and cost. Using low concentration of aldehydes (GA or FA), low uniformity of materials is obtained, but using high concentration, cytotoxic effect makes materials not suitable for biomedical application. Although protocols for removing unreacted GA were developed, GA using is still remains controversial. Despite being promising candidates, naturally crosslinkers like genipin do not present toxic effects, but the risk of instability of and low degree of crosslinking for crosslinked product reduces their potential use in biomedicine. EDC-NHS combination presented improved morphology and stability results as GA or genipin [25,26,27,28,29]. Physical crosslinking is a simple and safe technique and has the advantage of avoiding toxic effect of crosslinking agents. Physical crosslinking depends on many factors, such as radiation dosage, temperature, and hydration, and even if the crosslinking is obtained, an opposite process, UV denaturation, also appears [30]. It is also known that heating destroys the triple helical structure collagen and reduces its enzymatic resistance [11]. To avoid physical and chemical crosslinking disadvantages, using riboflavin as a chemical crosslinking agent and UV light results as GTA but without cytotoxic effects are obtained [31]. Enzymatic crosslinking presents excellent specificity and precise kinetics but it takes too much time and is expensive [27]. Blending is a very good compromise regarding the process and cost efficiency, in order to achieve desired and required properties according to the biomedical application. This can be also combined with chemical crosslinking, using a natural crosslinking agent or a lower quantity of a synthetic crosslinking agent, lowering the potential side effects. Blending method needs chemical structured, physical properties compatible materials in accordance with the desired potential application. Using blended biopolymers, the final materials present the benefits of all compounds, for example a blended collagen–cellulose hydrogel has also hemostatic and biomimetic properties from collagen and antibacterial and mechanical strength from cellulose. Collagen–chitosan blended hydrogel present immunogenic, carcinogenic, antifungal, and antibacterial properties [32,33,34,35,36,37].

The structure, sources, and properties such as pH, solubility, toxicity, stability, biomedical, and potential application of the most-used compounds of blended biopolymers are presented in Table 1.

## 3. Blended Collagen-Based Formulations—Synthesis, Characterization, and Properties

The traditional synthesis route for collagen hydrogels found in literature has two steps: Neutralization of acidic monomers with phosphate buffer (PBS) and NaOH solution and self-assembled process by warming solution to physiological temperature [36,37]. Starting from this route, there were developed different methods of synthesis for blended collagen-based hydrogels, as presented in Figure 2. However, the route, parameters, and compounds depend on required properties for potential application of final material. For example, for corneal tissue repair mechanical and optical properties were obtained using ion leaching techniques with NaCl and collagen I solution. Viscoelastic properties and transparency were also achieved when pH was increased to isoelectric point at a low ionic strength [38]. In order to obtain improved mechanical properties, greater pore sizes collagen hydrogels need to be synthesized at lower gelation temperatures due to the formation of fewer, longer, and thicker collagen fibrils important characteristics in drug delivery systems and wound healing. In tissue engineering application mimicking the specific morphology and manufacturing the 3D structures of cell are important parameters which need to be controlled using different methods: Realize the gelation process in customized micromolds or bioprinting technology [36,37,38,39].

There are many routes in literature for synthesizing blended collagen hydrogels and other formulations (collagen–alginate, collagen–chitosan, collagen–cellulose) with different crosslinking agents and parameters.

### 3.1. Blended Collagen–Alginate Biomaterials

Various synthesis methods are presented in literature for collagen–alginate hydrogels, with different parameters and physical forms (aerogel, films, gels) related to the application. Yang X. et al. and Bouhadir K.H. et al. mixed sodium alginate, sodium periodate aqueous solution, stirred for 4 h at room temperature in the dark, added ethylene glycol for half an hour, and purified by dialysis, removing excess of NaIO_4_ crosslinker, and then freeze-dryed. Using the swelling compressive modulus method, biodegradation, cell compatibility, and proliferation property were achieved for cartilage tissue engineering. However, in every synthesis, parameters should be changed in order to obtain the best combination of parameters properties for the desired application. In this case it was observed an increasing of degree of oxidation (DO) of alginate dialdehyde (ADA), produced slower degradation rate and increasing DO developed an increased compressive modulus [40,41].

#### 3.1.1. Synthesis and Characterization of Collagen–Alginate Aerogels

In order to synthesize a stable three-dimensional collagen–alginate aerogels, the obtained hydrogel as shown in Figure 3a were carried out two different processes: Water–solvent exchange and supercritical drying.

Synthesized collagen–alginate aerogels, as presented in Figure 3, have the ability to interact with cells, to develop attachment and proliferation. The obtained morphostructure is an interconnected porous network covered with a nonporous outer wall that remains chemically and biologically unchanged even after the supercritical drying process. Pores dimensions of 2–10 micrometers and 558.65 mg/mL density demonstrated to be proper for ta potential tissue engineering application [42].

#### 3.1.2. Synthesis and Characterization of Collagen–Alginate Films

Using route and parameters as described in Figure 4, homogenous collagen–alginate films were obtained. It was observed that for high collagen concentration, thermal stability increases, tensile strength decreases, and fluid uptake, ability of obtained films, improves up to 40%. Resulted collagen–alginate films presents high hemocompatible and no influence on rat mesenchymal stem cell (rMSC), proliferation, and viability [43].

#### 3.1.3. Synthesis and Characterization of Collagen–Alginate Hydrogels

Blended collagen–alginate gel synthesized as presented in Figure 5a, showed mechanical and structural improved properties and a good cell adhesion of encapsulated human induced pluripotent stem cells (iPSC)-derived neurons [44].

Other synthesis route for alginate–collagen hydrogel is presented in Figure 5b. CAF hydrogels presented pore sizes between 40 and 120 μm also exhibited improved mechanical and elastic properties and cytocompatibility for three cell lines: L929; hMSCs and MIN6 β-cells, with a microenvironment that supports cell survival and function for cell culture applications [45].

### 3.2. Blended Collagen–Chitosan Biomaterials

Collagen–chitosan is an important blend of biomaterials used in biomedicine, which presented interest for many research groups. In 2010, Wang et al. [46] used glycerolphosphate for a new chitosan–collagen blend with mimic proprieties of extracellular environment and direct cell function. A high expression of levels of osterix and bone sialoprotein genes in medium induced by the presence of chitosan was observed. This composite chitosan–collagen–hydroxyapatite with thermogelling properties was developed for bone tissues engineering. In 2010, Rao et al. [47] using Dylbecco’s modified Eagle’s culture medium studied the behavior of cell seeded and unseeded collagen–chitosan hydrogels. After 3 days of mineralization, the hydrogels became opaque. The obtained materials presented reduced cell viability in proliferation rate and slightly increased rheological properties. Sionkowska et al. [48] improved the swelling behavior of collagen–chitosan hydrogels by adding tannic acid. The decreasing of compressive modulus with the increasing the amount of tannic acid was observed. In 2016, Teng et al. [49] using collagen–chitosan–hydroxyapatite mixture and Tylingo et al. [50] using collagen–chitosan–gelatin prepared in an emulsion, uniform biomimetic soft composite microspheres with regular shape and 5–10 nm diameter range. There were also developed chitosan–polyvinyl alcohol-based soft composite with extended shelf life antioxidant properties given by molecular complex between gallic acid and cyclodextrin [51,52,53].

Classical route of synthesis for blended collagen–chitosan hydrogels uses acetic acid as solvent starting from chitosan and collagen solution in different concentrations and ratios regarding the mechanical properties and porosity needed in correlation with the application. The next steps are: centrifugation at low temperature for a few minutes (4°, 10 min) or stirring at room temperature for 1 h, freezing (−40 °C for at least 2 h), freeze-drying in a lyophilizer (−80 °C, 24 h). Stability of the synthesized materials is obtained using crosslinking treatment: Chemical with GA, tetraethyl orthosilicate (TEOS), physical in the oven for 24 h treatment at 105 °C or enzymatic using transglutaminase and temperature treatment [54].

Using solvent casting technique and TEOS as crosslinking agent, as presented in Figure 6a, collagen–chitosan homogenous hydrogel with smooth surface was obtained. This morfostructure achieved due to good compatibility of collagen, chitosan and crosslinking agent. Loading 5–30%wt caffeic acid (CA) into blended hydrogel, determined nonhomogeneous surface material with superior thermal stability. Swelling behavior reached to a high ratio due to amino acids, but when load CA this decreased. The degradation behavior is also influenced by CA loading, producing roughness and cracks on materials surface. CA was loaded for its antioxidant activity in order to develop delivery systems which demonstrated a quick release within the first hour and the stabilization of release mechanism appeared between 1 and 8 h. Even if CA was an obstacle for molecular chain monovalent, its antioxidant activity when loaded to collagen–chitosan hydrogels demonstrated to be achieved for potential pharmaceutical and cosmetic application [55]. Biopolymeric blended collagen-chitosan films were obtained by solvent evaporation from bended mixture containing silver nanoparticles with antimicrobial properties [56]. Thermosensitive collagen–chitosan hydrogels were synthesized using thermal gelation at physiological pH and temperature, as presented in Figure 6b. It was obtained a compact, stiff matrix with increased alkaline phosphate activity and calcium deposition in hydrogels pores developed in presence of hydrochloric acid. This obtained Ca/P ratio, close to hydroxyapatite’s, recommends collagen–chitosan–alkaline phosphatase for potential in biomedical applications [46]. 

For hydrogels obtained as presented in Figure 6c, characterization was performed using SEM, FTIR, mechanical test, porosity, degradation kinetics, equilibrium swelling measurement and mass variation, cytotoxicity test, RSC96 culture, primary Schwann cells evaluation, subcutaneous implantation experiment and histological analysis. Collagen hydrogel mechanical properties and degradation rate were improved by adding chitosan. Cytocompatibility, cytotoxicity and the capability to promote the attachment, migration and proliferation of Schwann cells were also demonstrated on collagen–chitosan scaffolds. In vivo behavior for scaffold implantation showed obviously modulated degradation behavior without causing any inflammatory reaction. Therefore, the developed collagen–chitosan scaffolds may be promising candidates for future application in peripheral nerve regeneration and may have potential to extend for other tissue engineering fields, including bone, tendon and muscle [57].

Other methods of synthesis for collagen–chitosan and collagen–chitosan–hyaluronic acid were performed by Gilarska A. et al. by mixing collagen and chitosan solution (50:50) in acetic acid and genipin solution in PBS buffer. Collagen–chitosan–hyaluronic acid hydrogels were formulated using the same procedure followed by the addition of hyaluronic acid solution in PBS buffer. All the prepared polymeric sols were vigorously vortexed and incubated at 37 °C until gel formation. Based on the rheological measurement it was concluded that the crosslinking process was completed after 60 min and the final gels were obtained. There were obtained compact structures with improved mechanism properties, stable materials that slowly degrades on enzymes activity. It has also been demonstrated cell adhesion and proliferation of MG-63 cell line [58].

### 3.3. Blended Collagen–Cellulose Biomaterials

Collagen–celulose blended hydrogel was recently studied in order to improve mechanical properties, elastic modulus, comptability, secretions absorbtion, cicatrization process, healing time, and frequency of changing of active wound dressing.

Due to collagen–cellulose hydrogen bonds interaction and the ability of BC fibrils to coat and penetrate collagen molecules, foam structured collagen–cellulose blended hydrogels were obtained (Figure 7). No changes in the crystal structure and improved thermal stability were achieved with the incorporation of collagen. It was also observed a big increase Young’s Modulus and tensile strength and a slight decrease of the elongation at break. Using 3T3 fibroblast cells for cell adhesion studies, after 48 h of incubation collagen–cellulose blended hydrogels were capable of forming cell adhesion and proliferation. The collagen–cellulose material presented much better cytocompatibility than pure BC. There was synthesized bioactive foam structured collagen–cellulose hydrogel with good cell adhesion and attachment with potential use for wound dressing or tissue-engineering scaffolds [59].

## 4. Biomedical Pharmaceutical Application for Blended Collagen-Biomaterials

### 4.1. Drug Delivery Systems Based on Blended Collagen-Hydrogels

Classical drug administration of therapeutic molecules at high doses for expected therapeutic effect usually produces significant and undesirable side effects. In order to obtain the same or improved therapeutic effects and efficacy, local delivery or targeted delivery with natural active principle loading has been investigated.

Blended collagen biomaterials improved most of the time for both support materials the properties needed in drug delivery systems such as specific pore size for active principle loading, fibril distribution, enzymatic degradation, and stability in human body, being able to be formulated in many forms for different applications, Figure 8.

Drug delivery systems are complex alternatives of classical drugs with improved release properties (time, localization, side effects) and involve many fields in two important domains: materials science (for material synthesis and characterization processes) and pharmaceutical science (for support materials-active principle compatibility and interaction, efficacy, and therapeutically effect characterization). Characterization should include also materials properties and carrier forms (microspheres, hydrogels or gels, films) and release properties (release time and mechanism model). For example, collagen chitosan blended homogenous hydrogel obtained with TEOS 2% crosslinking agent presented 8 h release for loaded caffeic acid. Caffeic acid loading in collagen–chitosan blend hydrogels presents high efficiency but also a challenge. CA, found that in the bark of Eucalyptus globulus is a hydroxycinnamic acid, an organic compound with phenolic and acrylic functional groups. CA presents high-quality antioxidant properties and is therefore used in athletic supplements to boots the performance and to decrease the fatigue, but is also used in weight loss and cancer prevention. Furthermore, in medical research CA is strongly used due to its action against oxidative stresses [60,61,62,63,64,65,66].

Collagen–chitosan film with growth factor used in wound healing presents a release time of 28 days, poly (lactic-co-glycolic acid) (PLGA)/collagen membrane with vancomycin, gentamicin, and lidocaine used for antibiotic activity presented 4, 3 and 2 weeks and collagen–chitosan sponge with dexamethasone used in oral mucositis, 10 h in PBS [67,68,69]. Microsphere collagen–bacterial cellulose mechanism model was BSA as the model protein, and employing quasi-primary, quasi-secondary, and Kannan–Sundaram intragranular diffusion models, zero-order, first-order, Higuchi and Korsmeyer–Peppas models [70].

Carrier form, support material components, active substances or drugs and the applications of each drug delivery system based on blended collagen-hydrogels are presented in Table 2.

### 4.2. Potential Application of Blended Collagen-Hydrogels in Tissue Repair and Engineering

Collagen proteins are an important compound in tissues and organs and influence cell expressions. Blended collagen-based hydrogels provide new materials with improved properties for both of used hydrogels.

Blended collagen-based hydrogels can be used for cardiovascular diseases, restorative dentistry, neural diseases, bone defects, wound healing, and tendon damages. Mechanical properties of arterial circulation systems are established by collagen. Blended collagen-based biomaterials were similar to natural collagen presenting improved mechanical properties, native strength, low immune response, low calcification properties, with potential for reconstruction of blood vessels (Figure 9).

Approaches to biological surgical implants or bone tissue engineering have changed lately due to the availability of biologic prothesis as alternatives to autologous tissue transfer and synthetic meshes. Blended collagen-based biomaterials with natural materials stimulate and activate cells, induce a cellular specific cellular response, produce tissue regeneration and restore the original functionality. Depending on the organ or body zone, tissues present different mechanical, electrical, physical, degradation, proliferation, and biologic characteristics which a single material, even natural, cannot mimic. Blending multiple natural materials similar properties with native tissue are obtained, producing cell growth. For cardiac tissue repair, alginate–collagen–polysaccharide scaffolds mimic native tissues.

Bone defects caused by trauma, tumors, and infections are healed with tissues substitutes capable of spontaneous bone healing without immunogenic response or viral transmission. Despite the poor mechanical properties of collagen and the fact that scaffolds must tolerate mechanical stress for optimal reconstruction of hard tissues defects, collagen is the best biological material used in scaffold synthesis. To obtain a similar composition and structure, native bone, osteogenesis, and mineralization are improved by incorporating inorganic bioactive minerals. Blended collagen-based biomaterials with potential use for bone tissue engineering with high potential in literature are: BC/collagen, collagen–chitosan–nano-hydroxyapatite, collagen–alginate–nano-silica, collagen–alginate/titanium oxide (TiO_2_), collagen–alginate–fibrin, collagen–alginate, collagen–chitosan, core-shell fibrous collagen–alginate, chitosan, collagen–bioactive glass, nanocellulose/collagen–apatite hydrogels, and collagen–chitosan–silver nanoparticle films [1,2,11,13,16,26].

Collagen–chitosan–alginate hydrogels loaded with curcumin promote diabetic wound healing with anti-inflammatory and antioxidant properties, reduce secretions and improve the safety and efficiency of healing process [81].

Tendon healing is slower than other tissues having a limited blood and nerve supply, therefore tissue engineering is an important procedure for major tendon and ligament injuries. Collagen– chitosan hydrogel presents the ability to promote the attachment, migration and proliferation of Schwann cells, modulated degradation behavior without inflammatory reaction developing potential application in bone, tendon, and muscle engineering [57].

An important factor in wound healing is discovering the balance between the composition of support material and growth factors, extracellular matrix proteins, and stem cells, which should successfully complete all three phases of wound healing (inflammatory, proliferative, and remodeling) and restore tissue to its pre-injured state. Stem cells are the engine of the repair process, being responsible of immune response, inflammation, tissue protection and reparative mechanisms. Most interesting stem cells are MSCs, iPCS researched in many types of wound healings. MSCs showed its benefits in reducing inflammation, accelerating growth and collagen production and demonstrated its actions in vivo surgery for cartilage repair. However, major injures are still an important burden for the patient, the healing process is hard and durable and sometimes current therapy fails. For all these patients, new therapies are necessary and wound healing new materials based on collagen play a significant role because collagen is one of the components (with proteoglycans and fibronectin) of the matrix which is produced in the proliferative phase of healing. Moreover, blended material collagen-based, collagen–alginate for infertility application, collagen–alginate–fibrin for pancreas tissue engineering, collagen–chitosan for different tissues, are already tested. These blended collagen-based biomaterials developed no inflammatory reactions, good adhesion and proliferations with important perspective for in vivo test and clinical research in order to find a standardization of clinical trial guidelines [93,94,95]. In this application spectra of tissue repair and engineering, different blended materials with various forms, microsphere, films, aerogels, fibers, sponge with collagen, alginate, chitosan, hydroxyapatite, and cellulose are presented in Table 3 [48,91,92].

## 5. Conclusions and Future Approaches

Biopolymers are an important class of soft materials: They are smart and suitable for various applications in biomedical field. Their capability to control their properties according to the desired potential application, to absorb and retain significant amount of water, without dissolving or losing their three-dimensional structure are basic properties when a material is used as in biomedicine. Collagen is a natural material, the main component of tendons, which attach muscle to bone and the connective tissue within and around muscles. Is a biomaterial with low immunogenicity, good in vivo absorption, synergism with other bioactive compounds and hemostatic properties frequently used for cardiac application, cosmetic surgery, bone grafts, tissue regeneration, reconstructive surgical, wound healing, and drug delivery system. The biggest advantage of a collagen hydrogel is that cells and bioactive components can be incorporated directly into it during the fabrication process.

In order to improve or to obtain the required properties for the purposed application, collagen is used as a compound in blended biomaterials. Moreover, natural blended biomaterials collagen–alginate, collagen–chitosan, and collagen–cellulose are nontoxic, non-immunogenic, non-carcinogenic, biocompatibility, bio-absorbability, antimicrobial, antibacterial, antifungal, anticoagulant, anti-tumor, hemostatic, and biomimetic. These properties are obtained using different routes with different crosslinking agents and parameters when synthesizing blended collagen-based materials.

Blended collagen-based hydrogels can be formulated in a variety of physical forms including films, microspheres, micro and nanoparticles, coatings, and aerogels that provide a large spectrum of potential applications: Bone tissue engineering, infertility, periodontal tissue regeneration, postsurgical peritoneal adhesion prevention soft tissue engineering (pancreas tissue engineering and musculoskeletal applications), wound dressing, peripheral nerve regeneration, wound healing and diabetic wound healing, cardiac tissue engineering, coating of cardiovascular prostheses, support for cellular growth, and controlled drug delivery systems.

Drug delivery systems formulate microspheres, hydrogels or gels, membranes, nanoparticles, aerogels, sponges, or film which can carry different classes of drugs, antibiotics (aminoglycosides, fluoroquinolone, anthracycline, and glycopeptide), nanobodies, nonsteroidal anti-inflammatory, calcium-channel blockers, local anesthetics, corticosteroids, chemotherapy drug, and active principle from different plants: Aloe-vera or turmeric (curcumin). All these associations are used for cancer chemotherapy, diabetic wound healing, myocardial infarction, skin regeneration, cardiac disease, antibiotic therapy, oral mucositis, and tissue regeneration.

Instead of using collagen alone, the blend of collagen with biocompatible biomaterials and certain methods, compounds, or technologies recently-developed may be the direction of future research. Higher resolution and improved biomimetic properties are important future perspectives for collagen-based blended biomaterials. For example, stereolithography can produce blended collagen-scaffolds with controlled pore size. Other improvement can be obtained developing ink-jet bioprinting and valve-jet bioprinting for blended collagen biomaterials, which are already tested for collagen. These can provide a good control on cell distribution in the scaffolds. New perspectives could focus on urinary tract repair which inquires tubular shape and elasticity, properties offered easily by blended collagen-based hydrogels.

In the presented large spectrum of potential and further applications, good in vitro properties show a class of interesting materials for researchers, that will lead to a growth of new commercial products based on blended collagen–alginate, collagen–chitosan, or collagen–cellulose which can improve the biomedical therapy without dangerous effects on the human health and environment. However, for all these applications a stable partnership of mixed teams of researchers and entrepreneurs is vital.

## Figures and Tables

**Figure 1 materials-13-05641-f001:**
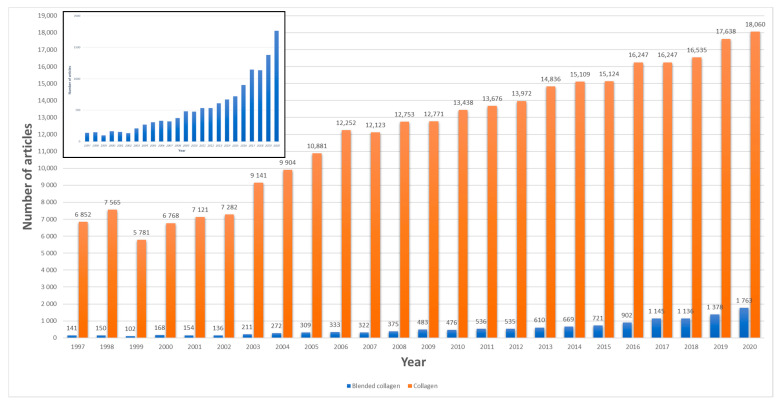
Annual publications on collagen and blended collagen 1997–2017. The search engine used Science direct, applying the terms “collagen” and “blended collagen”.

**Figure 2 materials-13-05641-f002:**
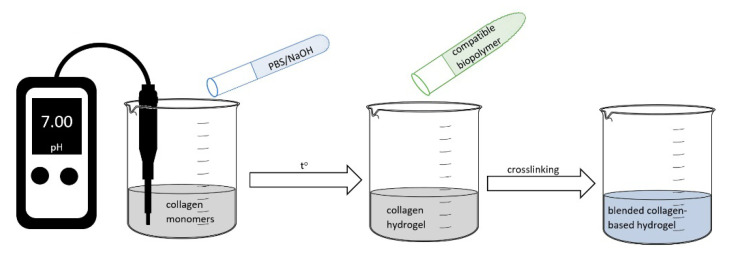
Graphical representation of traditional method for blended collagen-based hydrogels.

**Figure 3 materials-13-05641-f003:**
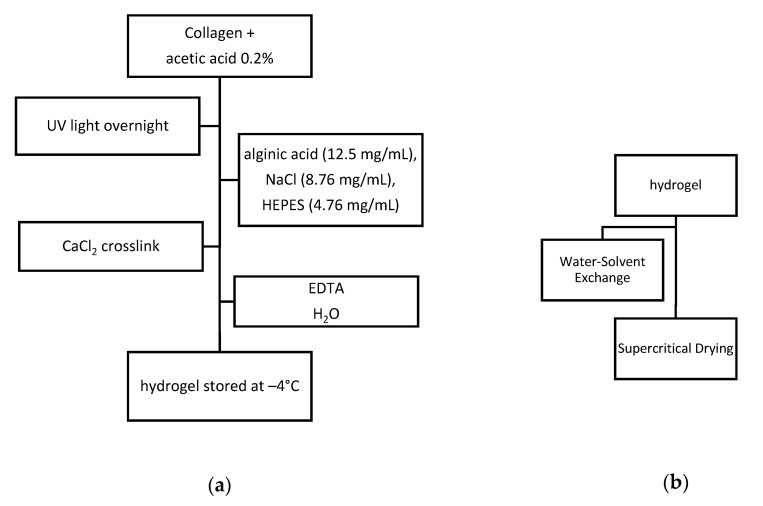
Synthesis process: (**a**) Synthesis of collagen–alginate hydrogel; (**b**) synthesis of aerogel (EDTA-ethylenediaminetetraacetic acid, HEPES-4-(2-hydroxyethyl)-1-piperazineethanesulfonic acid).

**Figure 4 materials-13-05641-f004:**
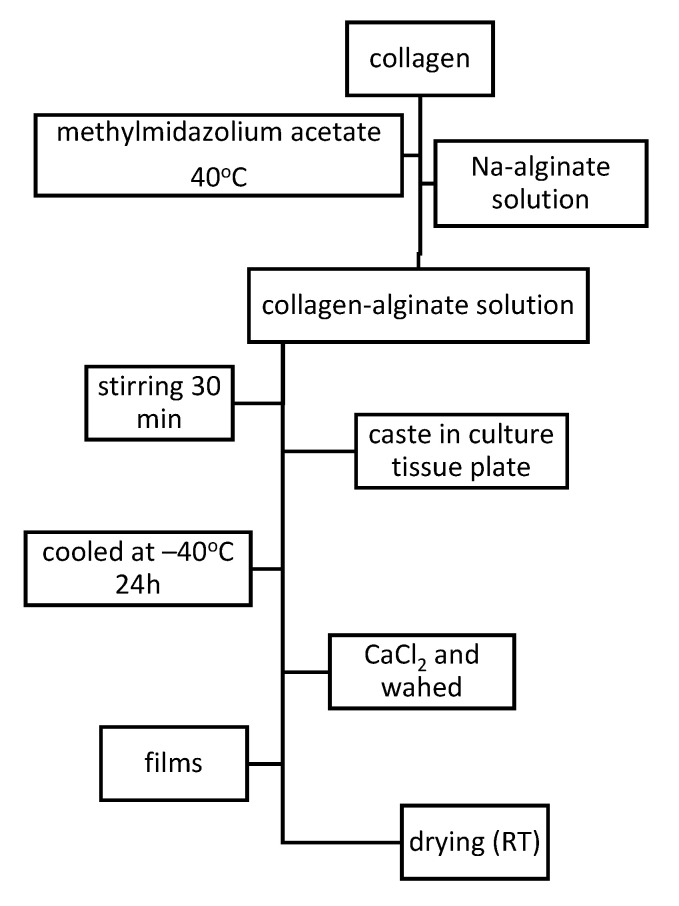
Synthesis of collagen–alginate films.

**Figure 5 materials-13-05641-f005:**
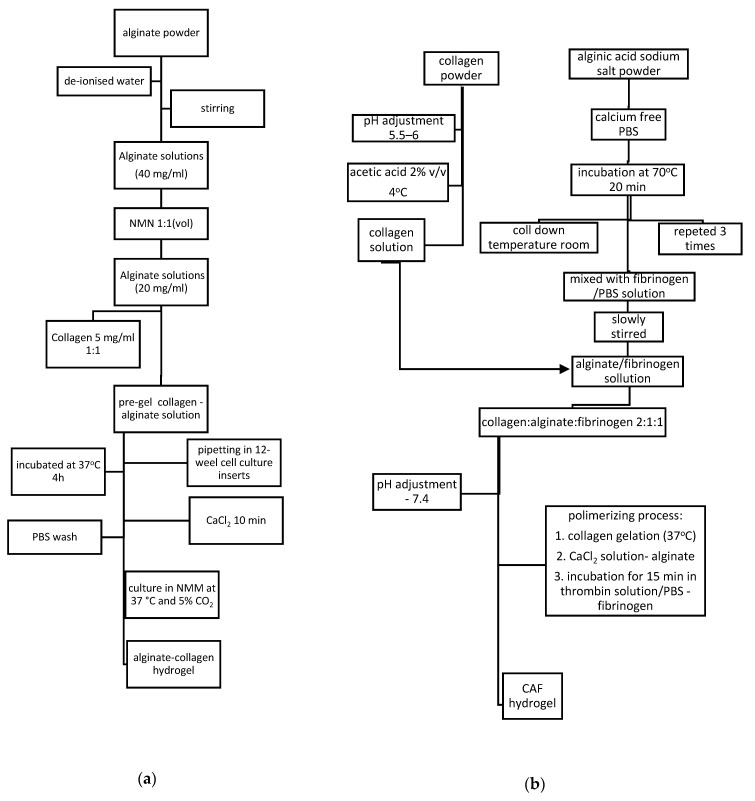
(**a**) Synthesis of collagen–alginate from pre-gel solution 20 mg/mL alginate and 5 mg/mL collagen. (**b**) Synthesis of collagen–alginate-fibrinogen (CAF) 2:1:1 hydrogel.

**Figure 6 materials-13-05641-f006:**
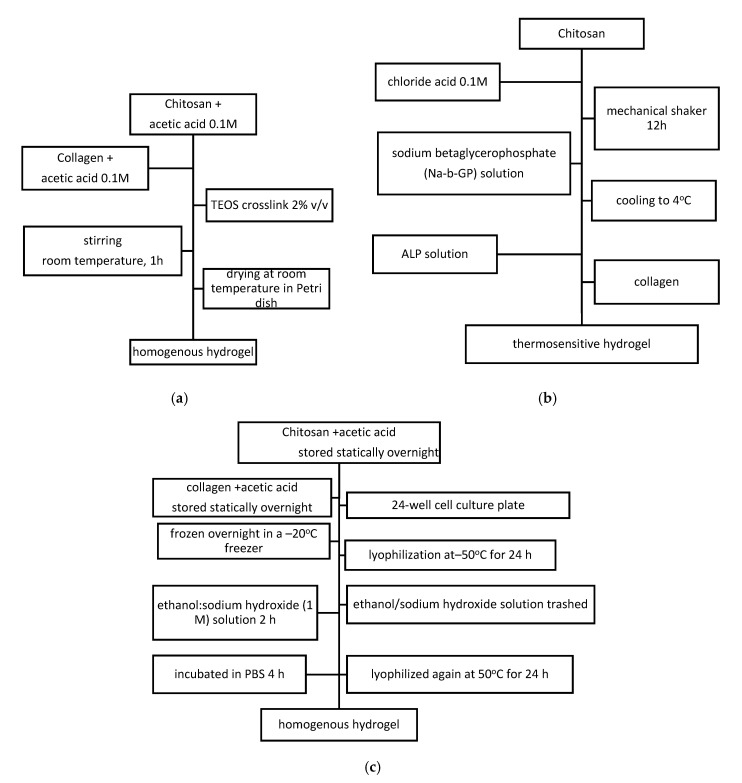
Synthesis route of: (**a**) Collagen–chitosan using pre-gel solution 10 mg/mL alginate and 2.5 mg/mL collagen. (**b**) Collagen–chitosan thermosensitive hydrogel. (**c**) Collagen–chitosan homogenous hydrogel for nerve tissue regeneration.

**Figure 7 materials-13-05641-f007:**
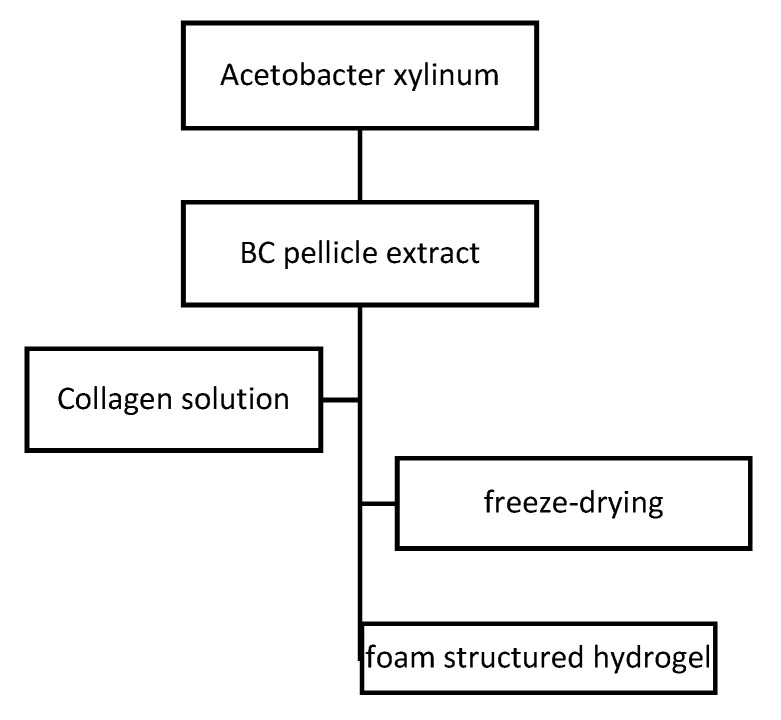
Synthesis route of bacterial cellulose (BC)/collagen hydrogel.

**Figure 8 materials-13-05641-f008:**
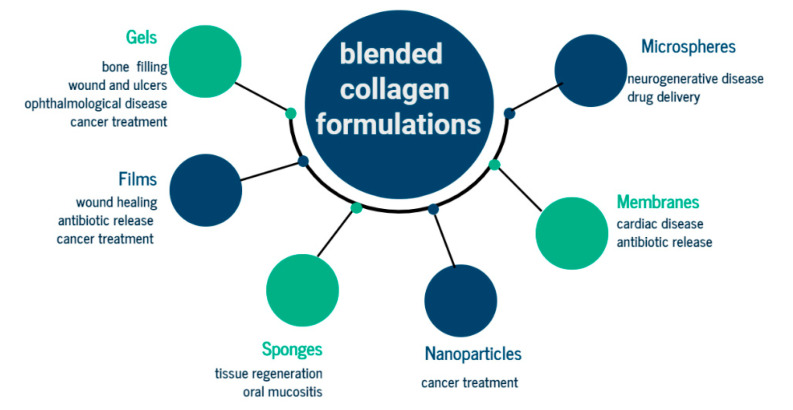
General biomedical application of drug delivery system based on blended collagen biomaterials.

**Figure 9 materials-13-05641-f009:**
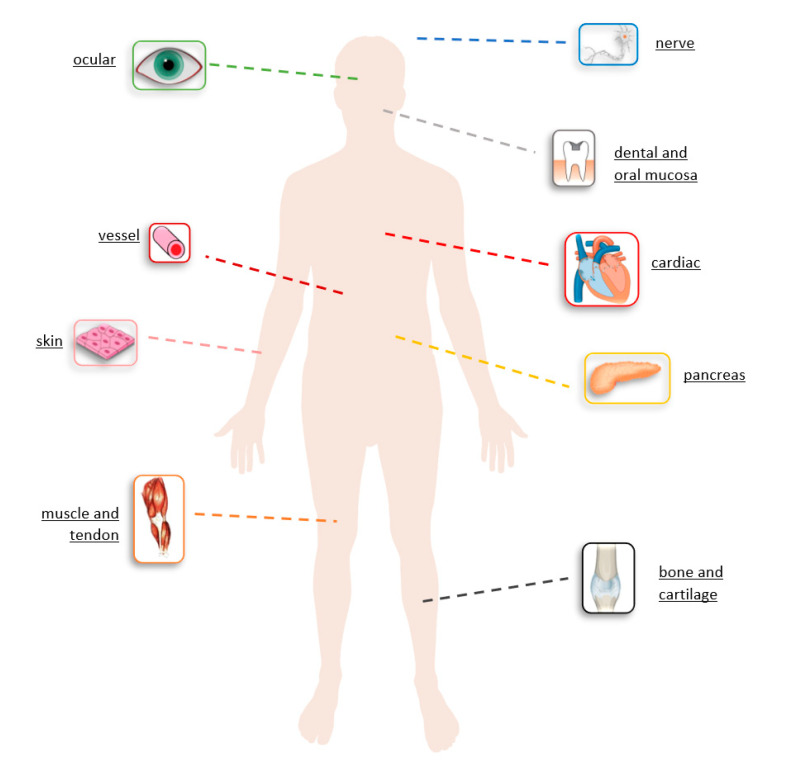
General biomedical application in tissue repair and engineering for different formulation of blended collagen biomaterials.

**Table 1 materials-13-05641-t001:** Biopolymers structures, sources, properties, and potential applications.

Biopolymers	Structure	Sources	Properties					Applications	Ref.
			Solubility	Toxicity	Ph	Stability	Biomedical		
Collagen	Triple helix, a unique tertiary structure three identical or non-identical polypeptide chains, each chain is composed of around 1000 amino acids or more in length	Vertebrate body protein (skin, tendon, bone, cornea, dentin, fibrocartilage, large vessels, intestine, uterus, dermis, tendon, placenta)	Insoluble in water, but lowering the pH of solution can increase solubility	Negligible	Solubility, structural, thermal properties are pH sensitive	Poor mechanical, thermal, enzymatic, tensile stiffness	Biodegradable, biocompatible, unique self-assembling fibril-forming properties, biomimetic	Food, cosmetic, photographic, ophthalmology, inserts, shields, particles, gels, aqueous injectables, drug delivery	[12,13,14,15,16,17,18,19,20,21,22,23,24,25,26,27,28,29,30,34]
Alginate	Whole family of linear copolymers containing blocks of (1,4)-linked β-D-mannuronate (M) and α-L-guluronate (G) residues	Brown algae (Phaeophyceae), Laminaria hyperborea, Laminaria digitata, Laminaria japonica, Ascophyllum nodosum, and Macrocystis pyrifera	High solubility in aqueous solutions lowering the pH of solution can increase solubility	Low	Viscosity of alginate solutions increases as pH decreases	Poor mechanical and chemical	Biocompatibility, mild gelation	Wound healing, delivery of bioactive agents such as small chemical drugs and proteins, and cell transplantation	[31]
Chitosan	Poly-(beta-1-4) N-acetyl-D-glucosamine reactive amino groups	Deacetylation of chitin from insects (cuticles), crustaceans (skeletons-crab, shrimp, lobster), cephalopod (exoskeleton)	Soluble in aqueous acidic media (higher 50% deacetylation) Solubility is enhanced by decreasing its molecular weight	Negligible	Solubility, biomedical properties are pH sensitive	-	Nontoxic, non-immunogenic, non-carcinogenic, biocompatibility, bio absorbability, antimicrobial, antibacterial, antifungal anticoagulant, anti-tumor, hemostatic	Biosensors, drug delivery, wound dressing	[32]
Cellulose	Exopolysaccharide	Wood, cotton, sugar beet, potato tubers, onion, hemp, flax, wheat straw, mulberry bark, algae, bacteria	-	Negligible	-	High mechanical	Biocompatibility, biodegradability, biological affinity, antibacterial	Wound dressing, shields, dental implants, artificial blood vessel	[33]

**Table 2 materials-13-05641-t002:** Drug delivery systems based on blended collagen-hydrogels.

Carrier Form	Support Material Component	Active Substance/Drug	Application	Ref.
Microsphere	Collagen–alginate	Glial cell line-derived neurotrophic factor	Neurodegenerative diseases	[71]
-	Collagen–bacterial cellulose	Bovine serum albumin	Potential drug delivery system	[70]
-	Collagen–chitosan–nano-hydroxyapatite		Potential drug delivery	[72]
Hydrogel or gel	Hydroxyapatite/collagen–alginate	Bone morphogenetic protein	Bone filler	[73]
-	Alginate–collagen	Doxycycline	Vision-threatening diseases	[74]
-	Collagen–carboxymethyl cellulose	Anti-inflammatory drug	Retinal ischaemia/reperfusion injury	[75]
-	Collagen–alginate	Methylene blue Imiquimod	Combinatorial photothermal and immuno tumor therapy	[76]
-	Collagen–chitosan	Nanobodies: 2D5 and KPU	Tumor treatment matrix for use in cancer therapy	[77]
-	Chitosan–alginate hydrogel encapsulated gelatin microspheres	5-fluorouracil	Anti-cancer drug delivery	[78]
-	Chitosan–collagen	Ibuprofen	Thermoresponsive scaffold	[79]
-	Collagen–chitosan–glucan	Aloe vera	Antibacterial activity against different types of bacteria (positive/negative grams) Wound healing Infected chronic wounds and ulcers	[80]
-	Chitosan–collagen–alginate	Curcumin	Diabetic wound healing	[81]
-	Chitosan–collagen	Glutanine-hstidme-arginine-glutamic acid-aspartic acid-glicine-serine (qhredgs)	Myocardial infarction	[82]
-	Collagen–chitosan	Norfloxacin	Wound healing, skin regeneration	[83]
Membrane	Collagen–chitosan	Nifedipine and propranolol hydrochloride	Cardiac disease	[84]
-	PLGA/collagen	Vancomycin, gentamicin and lidocaine	Antibiotic activity	[68]
Nanoparticle	Collagen–chitosan	Doxorubicin hydrochloride	Advanced cancer therapy	[85]
-	Gelatin-alginate/Fe_3_O_4_ magnetic nanoparticles	Doxorubicin hydrochloride	Cancer chemotherapy	[86]
Sponge	Collagen–chitosan	Dexamethasone	Oral mucositis	[69]
-	Collagen–PLGA	Gentamicin	Tissue regeneration	[87]
Film	Collagen–chitosan	Local anesthetics mix (lidocaine, tetracaine, and benzocaine)	Wound healing	[88]
-	Collagen–chitosan–hyaluronic acid	Gentamicin sulfate	Antibiotic release	[89]
-	Collagen–chitosan	Doxorubicin	Cancer treatment	[90]
-	Collagen–chitosan	Basic fibroblast growth factor	Wound healing	[67]
-	Collagen–chitosan–chondroitin	Loaded plga microspheres	Tissue engineering	[91]
-	Chitosan–collagen	Gentamicin sulfate	Antibiotic release	[92]

**Table 3 materials-13-05641-t003:** Tissue repair and engineering application.

Form	**Material Component**	Potential Application	Proprieties Improved/New	**Ref.**
Microsphereor bead	Bacterial cellulose and collagen	Bone tissue engineering	Adhesion, proliferation, and osteogenic differentiation	[96]
	Collagen–alginate	Infertility	Stem cells attachment, proliferation and differentiation	[97]
	Collagen–chitosan–nano-hydroxyapatite	Bone regeneration	High dispersity	[48]
	Collagen–alginate-nano-silica	Bone tissue engineering	Mechanical strength and generate porous membrane structure	[98]
Hydrogel or gel	Chitosan–collagen–gelatin	Wound healing	Rheology and mechanical properties	[50]
	Chitosan–collagen	Bone, tendon, muscle engineering	Mechanical properties, degradation rate, cytocompatibility, cytotoxicity and the capability to promote the attachment, migration and proliferation of Schwann cells; modulated degradation behavior without inflammatory reaction	[57]
	Chitosan–collagen	Cardiac cell culture and delivery	Thermoresponsive	[83]
	Collagen–sodium alginate-titanium oxide (tio_2_)	Periodontal tissue regeneration	Stiffness, water binding capacity, swelling, shrinkage factor, porosity and in-vitro biodegradation, osteocalcin secretion	[99]
	Collagen–alginate–fibrin	Soft tissue engineering (pancreas tissue engineering and musculoskeletal applications)	Thermo-responsive capacity at physiological conditions with stiffness similar to native soft tissues, enhanced osteogenic potential of human mesenchymal stem cells (hmscs) at high collagen content	[44]
	Gelatin–alginate–Laponite	Bone healing		[100]
	Collagen–chitosan	Wound dressing	Reduced swelling rate and improved mechanical strength, high cell survival rate and prominent spindle shape	[101]
	Chitosan–alginate hydrogel encapsulated gelatin microspheres - 5-fluorouracil	Soft tissue engineering anti-cancer drug delivery	-	[78]
	Chitosan–collagen	Tissue engineering	Modulate cell behavior	[102]
	Collagen–alginate- silver nanoparticles	Skin dressing	Antibacterial activity	[103]
	Collagen–chitosan–curcumin	Wound healing	-	[104]
	Chitosan–collagen	Peripheral nerve regeneration	Decreased the mean pore size, liquid uptake and degradation rate, increased the mechanical property of the composite scaffolds, good cytocompatibility without cytotoxicity	[105]
	Alginate–hyaluronic acid–collagen	Wound healing	Reepithelialization, collagen deposition, and angiogenesis in infected wound animal model	[106]
	Collagen–chitosan gel composite supplemented with a cell-penetrating peptide (CPP) (oligoarginine, r8)	Cutaneous wound healing Antibacterial activity	Inhibiting Staphylococcus aureus growth and had good ability to heal wounds	[107]
	Chitosan–collagen–alginate -curcumin	Diabetic wound healing	Anti-inflammatory and anti-oxidant, sustain drug carrier, wound healing, established wound healer as scaffold	[81]
	Chitosan–collagen–alginate	Wound healing	No significant cytotoxicity, and favorable hemocompatibility, biosecurity	[108]
	Chitosan–collagen	Bone and cartilage regeneration	Improved the mechanical properties, increasing the compressive strength, swelling ratio and prolonged the degradation rate	[109]
	Collagen–chitosan–hyaluronic acid	-	Improved mechanical properties and thermal stability	[110]
	Chitosan–collagen	Wound dressing	Gelation time, swelling behaviors, water evaporation rate and blood coagulation capacity	[111]
	Alginate–collagen	Neurogenesis and neuronal maturation	Improved mechanical properties	[43]
	Collagen–chitosan	Skin tissue engineering	L929 cell proliferation	[112]
	Alginate-gelatin-polysaccharide	Cardiac tissue engineering	Reduced water swelling, increased storage, modulus and loss modulus	[113]
	Chitosan–collagen	Scaffolds for tissue engineering	Morphology, mechanical stiffness, swelling, degradation and cytotoxicity	[114]
	Collagen–alginate	Tissue engineering	Mechanical support	[115]
	Core–shell fibrous collagen–alginate hydrogel	Bone tissue engineering	Viability, exhibiting significant cellular proliferation	[116]
	Cellulose–collagen	-	Morphology and thermal	[117]
	Collagen–chitosan	Skin tissue engineering	Accelerate cell infiltration and proliferation	[118]
	Collagen–chitosan	Cardiac repair	Promote cell migration	[119]
	Chitosan–collagen–bioactive glass	Regenerative medicine One-tissue bio applications	Thermosensitive	[120]
	Collagen–chitosan–hyaluronic acid	Bone tissue engineering	Osteosarcoma cell lines mg-63 cells adhesion, proliferation as well as alkaline phosphatase (alp) expression	[121]
	Collagen-alginate	Wound healing	Mechanical properties	[122]
	Poly (vinyl alcohol)–bioglass–chitosan–collagen	Bone tissue engineering	Mechanical, mineral deposition, biological properties and controlled release	[123]
	Collagen–chitosan–Au and Ag nanoparticles	Wound-healing	Antibacterial activity, good compatibility with living tissues	[124]
	Nanocellulose–collagen–apatite	Bone regeneration	No cytotoxic, genotoxic or mutagenic effects	[125]
	Chitosan–collagen–α, β-glycerophosphate	Tissue regeneration	Thermosensitive	[126]
	Chitosan–collagen matrix embedded with calcium-aluminate microparticles	Dental pulp stem cell	Biodegradation Differentiation of pulp cells	[127]
	Collagen–chitosan	Corneal tissue engineering	Optical properties, mechanical properties, suturability, permeability to glucose and albumin	[128]
Membrane	Collagen–chitosan	On-chip cell cultures Extracellular matrix supports	Mechanical properties and biodegradability	[129]
	Carboxymethyl chitosan–carboxymethyl cellulose–collagen	Postsurgical peritoneal adhesion prevention	Mechanical properties and biodegradability	[130]
Nanofibers	Collagen–cellulose	Tissue engineering.	Cell growth	[131]
	Collagen–chitosan	Biomedical applications	Mechanical properties	[132]
Aerogel Sponge/scaffold	Collagen–alginate	Tissue engineering	Cell migration, cell attachment, cell proliferation	[133]
	Collagen–chitosan	Tissue engineering	Thermostability Antibacterial activity	[134]
	Collagen–hydroxyapatite	Skeletal muscular system engineering	Mechanical strength Osteoinductivity	[135]
	Collagen–alginate	Cartilage, disc repair	Cell migration and proliferation	[136]
	Collagen–chitosan–polycaprolactone	Articular cartilage repair	Mechanical, swelling properties Growth of seeded chondrocytes	[137]
	Collagen–chitosan–poly (L-lactic acid-co-ϵ-caprolactone)	Vascular graft	Mechanical strength	[138]
	Collagen–alginate	Regenerative endodontics	Elastic modulus, tissue compaction and cell differentiation	[139]
	Collagen–chitosan–poly(L-lactide-co-glycolide)	Dermal tissue engineering	Mechanical properties Promote angiogenesis and induce in situ tissue formation	[140]
	Collagen–alginate–chitooligosaccharides	Skin tissue regeneration	Physicochemical, mechanical, biological properties	[141]
Film	Collagen–hydroxypropyl methylcellulose	-	Tensile strength, ultimate elongation and hydrophilicity of the blend film were superior to those of the pure collagen film Polyethylene glycol 1500 had almost no influence on the thermal properties of the blend film but obviously improved its stretch-ability and smoothness	[142]
	Collagen–chitosan–silver nanoparticles	Bone tissue engineering	Mechanical stability and antibacterial property	[143]
	Chitosan–collagen	Tissue engineering	Cell adhesion, morphological and biomechanical properties	[144]
	Collagen–chitosan	Coating of cardiovascular prostheses, support for cellular growth and in systems for controlled drug delivery	Dielectric permittivity, thermal stability, highest conductivity	[145]
	Collagen–chitosan–graphene oxide	Wound healing	Increased tensile, strength and brittleness, decreased elongation at break	[67]

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
