# Peer review of "Blended Natural Support Materials—Collagen Based Hydrogels Used in Biomedicine"

_materials, 2020, doi:10.3390/ma13245641_

Round 1

Reviewer 1 Report

The current manuscript entitled “Blended natural support materials -collagen-based hydrogels used in biomedicine” is designed and written well. Collagen is one of the well-known biomaterials and widely used for biomedical applications including different kinds of tissue construction. However, due to a lack of biophysical characteristics, it is often combined with other polymeric substances and ceramics and metal substances. In the current review, collagen combined with alginate, chitosan, cellulose to form hydrogels, film, and scaffold for biomedical applications. Further, the developed blended collagen hydrogels were applied for drug delivery systems and tissue engineering.

  1. The stem cell section can be included in the revised version of the manuscript.
  2. The quality of Figures 1 and 3 can be improved.
  3. The future approaches section can be included before the conclusion part.

Author Response

Thank you very much for your comments and observations, they help us to improve our manuscript. We have made the required changes.

  1. The stem cell section can be included in the revised version of the manuscript.

Since stem cells are used in wound repair, this is included in section 4.2. Potential application of blended collagen-biomaterials in tissue repair and engineering, line 335. However, paragraphs on stem cells were included in this section.

    2. The quality of Figures 1 and 3 can be improved.

Figure 1 and 3 quality have been improved, line189 and 241.

   3.The future approaches section can be included before the conclusion part.

Further approaches section was included in conclusion section, line 419.  

Reviewer 2 Report

This is an interesting review about discussing the significance of collagen based hydrogel. I have few minor points for consideration for the paper

1. Introduction has too many small paragraph, 1 or 3 liners. Kindly avoid giving such small separation. For e.g. Line 26 to 28, Line 34-35.

Also, the introduction is too short. Please use following reference for giving proper introduction to it

a) Use of collagen in tissue regeneration and nanodrug delivery: https://pubmed.ncbi.nlm.nih.gov/28796191/

b) collagen hydrogel in tissue regeneration: https://www.ncbi.nlm.nih.gov/pmc/articles/PMC4241868/#:~:text=Collagen%2Dbased%20hydrogels%20are%20gaining,natural%20extracellular%20matrix%20(ECM).&text=It%20is%20well%20known%20that,important%20role%20in%20cellular%20behavior.

2. Typos: Line 67 I think you mean "physical" crosslinking?  Moreover, line 67 to 69 makes no sense. Please modify it.

I think want to say physical crosslinking is not good method for gel formation as it does not provide necessary properties for desired biomedical application? In that case it will to be too vague to say this as most gels developed from natural biomaterials are physically crosslinked in nature.

Please either provide appropriate reference and specifically mention which biomedical application you are talking about?

3. Line 70, Enzymatic crosslinking is time consuming and costly affair. Please provide reference. 

4. Table 1. Has references for blending of collagen with natural biomaterial to get desired hydrogel. Please specify this in the legend.

5. Line 89. I can understand mechanical properties like viscoelastic nature and fibrous network of gels. But I am confused with what is optical property of hydrogel? Please expand on it if possible. And line 90 please provide isoelectric point of collagen if possible.

6. Please provide abbreviation as they first appear in the text for instance line 106, ADA; line 107 DO ADA? Entire text is without appropriate without full-form. Kindly add full-forms for acronyms as they appear first.

7. Figure 1b: Why hydrogel (a) in the box? is it typo? Kindly check such errors through out manuscript for e.g. table 2 what is QHREDGS? and in table 3 title "proprieties improved".. please check them and correct it appropriately.

8. Please avoid using single word subtitles. For example Aerogel, Film.. instead please use something like "synthesis and characterisation of collagen composite aerogel" or film etc.

9. Title of the work is hydrogel. But authors have explained significance of collagen alginate films, microspheres and other formulations. I am not sure what is the correlation of between films and hydrogels? I think authors should change title to more broader category to incorporate various solid and semisolid nanoformulations of collagen.

10.  Entire article have discussed FTIR, SEM and other techniques for charachterisation of formulations. If possible please provide images data by taking appropriate copyright form. It is useful to show the particle size data or morphological information from technical analysis. In-case of non availability please avoid mentioning direct information of SEM data shows or FTIR analysis shows, etc. Instead only focus on the results, without mentioning technique, instead mention particle size study or chemical interaction study etc.

11. Line 336 and 337. Indeed gels are better matrix to deliver cells and bioactive compounds. But to my understanding you have not explained any example of this in your main text. Please clarify and add appropriate data for supporting this concluding statement.

12. Line 344 and 350. I am confused you are talking about gels and moving to formulations like microspheres, micro and nanoparticles, coatings and aerogels?. These formulations cannot be considered as physical forms of hydrogels are they are different formulations with distinct mode of deliveries and physicochemical behaviours. They are not gels.

Are you trying to say hybrid-hydrogels with nanoparticles, microspheres and other formulations in collagen gel matrix?  Or are you individually talking about independent collagen based nano or micro-formulations? Kindly clarify and modify sentence appropriately in the text.

Author Response

Thank you very much for your comments and observations, they help us to improve our manuscript. We have made the required changes.

  1. Introduction has too many small paragraph, 1 or 3 liners. Kindly avoid giving such small separation. For e.g. Line 26 to 28, Line 34-35.

Also, the introduction is too short. Please use following reference for giving proper introduction to it

  1. a) Use of collagen in tissue regeneration and nanodrug delivery: https://pubmed.ncbi.nlm.nih.gov/28796191/
  2. b) collagen hydrogel in tissue regeneration:

https://www.ncbi.nlm.nih.gov/pmc/articles/PMC4241868/#:~:text=Collagen%2Dbased%20hydrogels%20are%20gaining,natural%20extracellular%20matrix%20(ECM).&text=It%20is%20well%20known%20that,important%20role%20in%20cellular%20behavior.

Introduction has been significantly improved, including more refences and information about collagen-based formulations.

  1. Typos: Line 67 I think you mean "physical" crosslinking?  Moreover, line 67 to 69 makes no sense. Please modify it.

I think want to say physical crosslinking is not good method for gel formation as it does not provide necessary properties for desired biomedical application? In that case it will to be too vague to say this as most gels developed from natural biomaterials are physically crosslinked in nature.

Please either provide appropriate reference and specifically mention which biomedical application you are talking about?

Physical crosslinking for collagen-based materials produces structure denaturation, the triple helical structure of collagen can be destroyed and enzymatic resistance is reduced.

3.Line 70, Enzymatic crosslinking is time consuming and costly affair. Please provide reference. 

References were restructured.

4.Table 1. Has references for blending of collagen with natural biomaterial to get desired hydrogel. Please specify this in the legend.

Please be more specific.

  1. Line 89. I can understand mechanical properties like viscoelastic nature and fibrous network of gels. But I am confused with what is optical property of hydrogel? Please expand on it if possible. And line 90 please provide isoelectric point of collagen if possible.

Optical properties were attributed to potential application of synthesized material in corneal disease. It was mentioned in the text, line 159.

  1. Please provide abbreviation as they first appear in the text for instance line 106, ADA; line 107 DO ADA? Entire text is without appropriate without full-form. Kindly add full-forms for acronyms as they appear first.

The text was fully re-read and acronyms problem was solved.

  1. Figure 1b: Why hydrogel (a) in the box? is it typo? Kindly check such errors through out manuscript for e.g. table 2 what is QHREDGS? and in table 3 title "proprieties improved".. please check them and correct it appropriately.

Entire text was checked.

  1. Please avoid using single word subtitles. For example Aerogel, Film.. instead please use something like "synthesis and characterisation of collagen composite aerogel" or film etc.

Every subtitle was changed.

  1. Title of the work is hydrogel. But authors have explained significance of collagen alginate films, microspheres and other formulations. I am not sure what is the correlation of between films and hydrogels? I think authors should change title to more broader category to incorporate various solid and semisolid nanoformulations of collagen.

Yes, the paper is about hydrogels that can be formulated in different physical forms. https://doi.org/10.1016/j.polymer.2008.01.027

  1. Entire article have discussed FTIR, SEM and other techniques for charachterisation of formulations. If possible please provide images data by taking appropriate copyright form. It is useful to show the particle size data or morphological information from technical analysis. In-case of non availability please avoid mentioning direct information of SEM data shows or FTIR analysis shows, etc. Instead only focus on the results, without mentioning technique, instead mention particle size study or chemical interaction study etc.

Figures for discussed technique were asked, but without any response. Meanwhile each paragraph was reformulated.

  1. Line 336 and 337. Indeed, gels are better matrix to deliver cells and bioactive compounds. But to my understanding you have not explained any example of this in your main text. Please clarify and add appropriate data for supporting this concluding statement.

More information about hydrogels were included in section 2.

  1. Line 344 and 350. I am confused you are talking about gels and moving to formulations like microspheres, micro and nanoparticles, coatings and aerogels?. These formulations cannot be considered as physical forms of hydrogels are they are different formulations with distinct mode of deliveries and physicochemical behaviours. They are not gels.

Are you trying to say hybrid-hydrogels with nanoparticles, microspheres and other formulations in collagen gel matrix?  Or are you individually talking about independent collagen based nano or micro-formulations? Kindly clarify and modify sentence appropriately in the text.

Paragraph was modified.

Reviewer 3 Report

Authors reviewed collagen-based hydrogels for biomedicine applications, however the interst of the present review is very limited, being only one section of interred for the readers.

Abstract. It must be completely rewritten. First the beginning is to basic and the interest of the review and the advance with respect of the state of art is not clear. Which is the different between this work and the thousands of similar reviews. Please, try to give value to your work.

Introduction. What is the point? It is so general that do not have a clear idea of the interest. There are only few references.

  1. Hydrogels structure and properties.

The information in this section is extremely basic in the first two paragraph and then, in the following is rather incomplete some examples of crosslinkers are given but the main crosslinkers are not described.

Section 3.

The Figures are not very useful. Instead of given the name of the different analysis performed, the synthetic pathway and the summary of the properties of the obtained hydrogels in each example must be more interesting for the readers.

Section 4,

I believe that the part of the different applications must be the core of this review, the other part are almost meaningless. In my opinion this section must be extended.  

Author Response

Thank you very much for your comments and observations, they help us to improve our manuscript. We have made the required changes.

Authors reviewed collagen-based hydrogels for biomedicine applications, however the interst of the present review is very limited, being only one section of interred for the readers.

Collagen based materials in biomedical application field is an inserting subject for researchers, due to the statistical data exposed in introduction section. It can be observed an increasing number of articles, yearly.

Abstract. It must be completely rewritten. First the beginning is to basic and the interest of the review and the advance with respect of the state of art is not clear. Which is the different between this work and the thousands of similar reviews. Please, try to give value to your work.

Introduction. What is the point? It is so general that do not have a clear idea of the interest. There are only few references.

Introduction section has been improved.

Hydrogels structure and properties.

The information in this section is extremely basic in the first two paragraph and then, in the following is rather incomplete some examples of crosslinkers are given but the main crosslinkers are not described.

Crosslink is a very discussed subject in reviews, it was only mentioned in here just to remind its importance. However, this section was improved.

Section 3.

The Figures are not very useful. Instead of given the name of the different analysis performed, the synthetic pathway and the summary of the properties of the obtained hydrogels in each example must be more interesting for the readers.

Each section was modified.

Section 4,

I believe that the part of the different applications must be the core of this review, the other part are almost meaningless. In my opinion this section must be extended.  

The section had been extended.

Round 2

Reviewer 3 Report

The manuscript have been significantly improved